# Anticancer Effect of Cathelicidin LL-37, Protegrin PG-1, Nerve Growth Factor NGF, and Temozolomide: Impact on the Mitochondrial Metabolism, Clonogenic Potential, and Migration of Human U251 Glioma Cells

**DOI:** 10.3390/molecules27154988

**Published:** 2022-08-05

**Authors:** Alexandr N. Chernov, Tatiana A. Filatenkova, Ruslan I. Glushakov, Alexandra S. Buntovskaya, Diana A. Alaverdian, Anna N. Tsapieva, Alexandr V. Kim, Evgeniy V. Fedorov, Sofia S. Skliar, Marina V. Matsko, Elvira S. Galimova, Olga V. Shamova

**Affiliations:** 1Scientific and Educational Center “Molecular Bases of Interaction of Microorganisms and Human”, World-Class Research Center “Center for Personalized Medicine”, Institute of Experimental Medicine, 197376 Saint-Petersburg, Russia; 2Department of Pharmacology with a Course in Clinical Pharmacology and Pharmacoeconomics, St. Petersburg State Pediatric Medical University, 194100 Saint-Petersburg, Russia; 3Department of Medical Biotechnologies, University of Siena, 53100 Siena, Italy; 4Children’s Neurosurgical Department No.7, Almazov Medical Research Centre, 197341 Saint-Petersburg, Russia; 5Laboratory of Neurooncology of Polenov Neurosurgical Institute, Almazov National Medical Research Centre, 197341 Saint-Petersburg, Russia; 6Department of Oncology, Medical and Social Institute, Saint-Petersburg University, 199034 Saint-Petersburg, Russia; 7Interdisciplinary Laboratory for Neurobiology, Sechenov Institute of Evolutionary Physiology and Biochemistry, Russian Academy of Sciences, 194223 Saint-Petersburg, Russia

**Keywords:** human glioma U251, cathelicidin LL-37, protegrin PG-1, nerve growth factor NGF, temozolomide, migration, clonogenicity, metabolism of mitochondria, OCR, ECAR

## Abstract

Glioblastoma (GBM) is one of the most aggressive and lethal malignancy of the central nervous system. Temozolomide is the standard of care for gliomas, frequently results in resistance to drug and tumor recurrence. Therefore, further research is required for the development of effective drugs in order to guarantee specific treatments to succeed. The aim of current study was to investigate the effects of nerve growth factor (NGF), human cathelicidin (LL-37), protegrin-1 (PG-1), and temozolomide on bioenergetic function of mitochondria, clonogenicity, and migration of human U251 glioma cells. Colony formation assay was used to test the ability of the glioma cells to form colonies in vitro. The U251 glioma cells migration was evaluated using wound-healing assay. To study the mitochondrial metabolism in glioma cells we measured oxygen consumption rates (OCR) and extracellular acidification rates (ECAR) using a Seahorse XF cell Mito stress test kit and Seahorse XF cell Glycolysis stress kit, respectively. We revealed that LL-37, NGF, and TMZ show strong anti-tumorigenic activity on GMB. LL-37 (4 μM), TMZ (155 μM), and NGF (7.55 × 10^−3^ μM) inhibited 43.9%–60.3%, 73.5%–81.3%, 66.2% the clonogenicity of glioma U251 cells for 1–2 days, respectively. LL-37 (4 μM), and NGF (7.55 × 10^−3^ μM) inhibited the migration of U251 glioma cells on the third and fourth days. TMZ also inhibited the migration of human glioma U251 cells over 1–3 days. In contrast, PG-1 (16 μM) stimulated the migration of U251 glioma cells on the second, fourth, and sixth days. Anti-mitogenic and anti-migration activities of NGF, LL-37, and TMZ maybe are relation to their capacity to reduce the basal OCR, ATP-synthetase, and maximal respiration of mitochondria in human glioma U251 cells. Glycolysis, glycolytic capacity and glycolytic spare in glioma U251 cells haven`t been changed under the effect of NGF, LL-37, PG-1, and TMZ in regard to control level. Thus, LL-37 and NGF inhibit migration and clonogenicity of U251 glioma cells, which may indicate that these compounds have anti-mitogenic and anti-migration effects on human glioma cells. The study of the mechanisms of these effects may contribute in the future to the use of NGF and LL-37 as therapeutic agents for gliomas.

## 1. Introduction

There has been a steady growth in morbidity and mortality associated with brain tumors worldwide in the last decade [1]. Brain tumors are a heterogenic group that consists of more than 100 histological subtypes [2]. The most frequently malignant brain tumors belong to glioblastoma (GBM) and anaplastic astrocytoma (AA) in adults and medulloblastoma (MB) in children [3]. These patients are subjected to complex therapy including surgical resection, radiotherapy, and chemotherapy. However, the efficacy of such therapy is low. The five-year survival rate of GBM patients reaches only 5.1% [4].

The basic reason for the low efficacy of therapy for brain tumors is the heterogeneity of cells in tumors and the selection of neoplastic clones in response to the development of drug resistance to the effects of different therapeutic modalities [5]. The reason of it is accumulation of somatic driver mutations in G2-M DNA checkpoint and different (Notch, NF-κB, Wnt/β-catenin, PI3K/Akt) signaling cascades of stem cells stimulate the development of the tumor [6,7,8,9]. On the other hand, the tumor microenvironment (astrocyte, epemdymocytes, microglial cells) secretes growth factors, cytokines, and peptides, which maintain of proliferation, migration, and angiogenesis of the tumor cells and preserve their from hypoxia, acidosis, impact of chemotherapy, radiation, generation of reactive oxygen species (ROS), and immune evasion [6,10]. In gliomagenesis and under the radiation, chemotherapy treatment the generation of ROS and oxidative damage of the cells was occurred [6,11]. These processes are connected with cell apoptosis, bioenergetic and respiratory capacity of mitochondria in cancer cells [12]. Furthermore, in low concentration ROS participates in an intracellular signal transduction, gene expression and stimulate cell proliferation. In high concentration ROS induce an apoptotic cell death via calcium overload in mitochondria, endoplasmic reticulum stress and the accumulation of aggregated proteins [13,14]. Bioenergetic and respiratory capacity of mitochondria is connected with the generation of ATP in a glycolysis and mitochondrial respiration. Consequently, these events in neoplastic cells intensify their prolific activity and migration, which is observed in the frequent recurrence and metastasis of neoplasia [15].

Therefore, there is an urgent need to search for new and more efficient therapeutic drugs and to study the proliferation and migration effects of tumor cells.

Regulatory molecules participate in many biochemical and physiological processes, such as growth factors and peptides of innate immunity. In the brain, among growth factors, the key role belongs to nerve growth factor (NGF). Thus, this growth factor presents the most promise for the treatment of nervous diseases. NGF binds to two receptors: p75 and TrkA [16]. p75 (NGFR) is a low-affinity receptor for some neurotrophins (NGF, BDNF, NT3, NT4). Neurotrophic receptor tyrosine kinase 1 (NTRK1 or TrkA) represents a high-affinity receptor of NGF [16]. The interaction of NGF with the TrkA receptor promotes the viability and differentiation of sympathetic cholinergic neurons and glial cells [17]. NGF, through the TrkA receptor, stimulates the migration and invasion of breast cancer MDA-MB-231 and MDA-MB-453 cells [18]. At the same time, NGF, through p75, induces cell apoptosis [17]. Researchers have studied the anticancer effects of NGF in regard to some tumor types. In the presence of NGF, medullary pheochromocytoma and prostate cancer cells stop their division and transform into neuron-like cells, with the inhibition of DNA replication, the appearance of processes, the formation of pseudogangia and the electrical excitability of the membrane [19]. Similarly, experiments show that NGF inhibits the proliferation and stimulates the neurogenesis of PC12 pheochromocytoma cells [20] and induces the autophagy of Schwann cells [21].

To date, more than 23,253 antimicrobial peptides (AMP) have been identified [22]. Most of the AMPs are molecules consisting of 12–50 amino acids, with a high content of arginine and/or lysine. Although, primarily, the peptides were considered antimicrobial, as compounds possessing antibiotic activity in regard to bacteria, unicellular fungi, protozoa and viruses, it was found that some peptides demonstrate pronounced cytotoxic activity against neoplastic cells [23]. Peptides have different structures. The presence and intensity of the effect depend on the structural characteristics of each peptide. In order to study the anticancer mechanisms, we evaluated two peptides of the cathelicidin family with different structures: cathelicidin (LL-37) with an α-helical structure, from the azurophilic granules of human neutrophils, and protegrin-1 (PG-1) with a β-hairpin conformation, from pig neutrophils.

Cathelicidin LL-37 shows tissue-specific action in regard to the cells of different types of tumors [24]. In previous studies, LL-37 was detected in the cells of lung epithelial adenocarcinoma (A549), epithelioid carcinoma (A431), Epstein–Barr virus (EBV)-transformed B cells, acute myeloid leukemia (HL-60, MG63), erythromyeloid leukemia (K562), lymphoma (U937), hepatoma (Hep22a), colon cancer epithelial cells (HT-29, HCT116), oral squamous cell carcinoma, breast cancer MCF-7, MDA-MB-435s and MDA-MB-231 cells, the pancreas, the prostate, A375 and A875 melanoma cells, the stomach, the ovaries and SH-SY5Y neuroblastoma cells [25]. The mechanisms of LL-37 involve interactions with FPR2, CXCR2, P2Y11, P2X7, MrgX2, EGFR/ErbB1, ERBb2, IGF1R, LGIC and TLR receptors, which are significantly expressed in different types of tumors as compared with normal cells. For example, a rise in the expression of the CAMP gene and levels of LL-37 secretion are associated with the progression of lung adenocarcinoma, breast, pancreatic and prostate cancers, ovarian cancer, melanoma and squamous cell skin cancer, wherein LL-37, through the CXCR4 receptor, stimulates the migration of breast cancer cells [26]. In contrast, the expression of the CAMP gene and levels of LL-37 secretion are significantly decreased in gastric and colorectal cancers, oral squamous cell carcinoma, leukemia, lymphoma and SH-SY5Y neuroblastoma cells [25].

The mechanisms of PG-1 in regard to cancer cells are still poorly studied. Experimental data show that protegrin-1 damages membranes. Moreover, PG-1 displays cytotoxic effects towards human cancer cells (breast carcinoma MCF-7, erythromyeloid leukemia K562, histiocytic lymphoma U-937, epithelioid lung carcinoma A-549, epidermoid carcinoma A-431, osteosarcoma MG-63 and doxorubicin-resistant cells). PG-1 and LL-37 synergically promote the anticancer effects of chemotherapy, damaging cell membranes and/or penetrating into the intracellular space, showing a more expressed effect on tumor cells than healthy cells. In high concentrations, these peptides are toxic to human cells [27], which creates an obstacle for their implication into medical practice.

Thus the aim of the current study was to evaluate the effects of nerve growth factor (NGF), human cathelicidin LL-37, protegrin-1 (PG-1), and temozolomide (TMZ) on the human U251 glioma cells in terms of bioenergetic function of mitochondria, clonogenicity, and migration rate.

## 2. Results

### 2.1. Effect of LL-37, PG-1, NGF, and TMZ on the Clonogenicity of Human Glioma U251 Cells

We studied the effects of LL-37, PG-1, NGF, and TMZ on the clonogenicity of human glioma U251 cells over 1–7 days (Table 1, Figure 1).

The data in Table 1, Figure 1B demonstrate that LL-37 (4.0 μM) inhibited 60.3% (*p* = 0.05) и 43.9% (*p* = 0.0001) of the clonogenic capability of human glioma U251 cells over 1 day and 2 days, respectively, wherein the percentage of single cells was strongly increased (66.7% and 35.3%) over 1 day and 2 days, respectively, as compared to the control. Increased counts of single cells were preserved 3rd and 6th days after the effect of LL-37. Similarly, NGF (100 μg/mL, 7.55 × 10^−3^ μM) also inhibited 66.2% (*p* = 0.0152) of the clonogenic capability of human glioma U251 cells over 1 day (Figure 1D). It promoted up to 60.0% higher number of single cells in the field of view. The impact of NGF was decreased, and the proliferation activity of glioma cells was increased in 2 days, wherein we observed a high percentage of single cells (20%). PG-1 (16.0 μM) did not inhibit the clonogenic capability of human glioma U251 cells over 1-7 days (Figure 1C,H,M). On the second day, the percentage of single cells (27.3%, in the control 8.6%) was increased. TMZ (155 μM) also inhibited 81.3%, 73.5% and 74.2% (*p* = 0.001) of the clonogenic capability of human glioma U251 cells over 1 day, 2 day and 3 day (Figure 1E,J,O).

### 2.2. Influence of LL-37, PG-1, NGF, and TMZ on the Respiratory Capacity of Mitochondria in Human Glioma U251 Cells

In order to study the possible mechanisms of action of NGF, LL-37, PG-1, and TMZ on the proliferative activity of U251 glioma cells, we measured of the oxygen consumption rate (OCR) and the respiratory capacity of mitochondria of these cells under the 2 h effect of NGF (7.55 × 10^−3^ µM), PG-1 (16 µM), LL-37 (4 µM) and temozolomide (155 µM) by XF Cell MitoStress test kit and Extracellular Flux Analyzer as previously described (Figure 2). Oligomycin, carbonyl cyanide p-trifluoromethoxyphenylhydrazone FCCP, rotenone, and antimycin A were used for detection of the level of non-mitochondrial respiration.

The data presented in Figure 2 show that NGF, LL-37, PG-1 and TMZ decreased the basal oxygen respiration (*p* = 0.0027), (*p* < 0.0001), (*p* = 0.0008), (*p* < 0.0001) respectively. NGF, LL-37, and TMZ have inhibited the ATP-linked oxygen consumption (*p* < 0.0001). NGF, LL-37, PG-1 and TMZ have also reduced a maximal respiration capacity (*p* = 0.0047, *p* = 0.0004, *p* = 0.004 and *p* = 0.0007 respectively) and activity of non-mitochondrial (cytosolic) oxygenases (*p* = 0.0008, *p* < 0.0001, *p* = 0.0001 and *p* = 0.0004 respectively) in regard to control. Spare respiration capacity hasn’t changed. TMZ and LL-37 have also reduced (*p* = 0.0003, *p* < 0.0001) of basal respiration capacity from NGF, PG-1 respectively. The influence of TMZ was lower (*p* < 0.0001, *p* = 0.0420) of ATP production in regard to PG-1 and NGF respectively. LL-37 has also inhibited of ATP production (*p* = 0.0006) in comparison with PG-1. Also TMZ has inhibited of a maximal respiration capacity (*p* < 0.0001) in regard to PG-1 and NGF. LL-37 has also reduced of a maximal respiration capacity (*p* < 0.0001) in comparison with NGF, PG-1), but has stimulated it (*p* < 0.0001) in regard to TMZ. PG-1 has stimulated of ATP production (*p* = 0.0007) in regard to NGF.

Using a XF Cell MitoStress test kit we also measured extracellular acidification rate (ECAR) and pH of medium and estimated a glycolytic state of mitochondria under 2 h effect of NGF (7.55 × 10^−3^ µM), PG-1 (16 µM), LL-37 (4 µM) and temozolomide (155 µM), Figure 3.

The data presented in Figure 3 demonstrate that TMZ (155 µM) and LL-37 (4 µM) decrease (*p* = 0.0043, and *p* = 0.0022) of extracellular acidification rate for a non-mitochondrial respiration in regard to control. PG-1, TMZ and LL-37 inhibited ECAR for ATP production in regard to control. TMZ also reduced ECAR for maximal respiration from control. PG-1, LL-37 and TMZ inhibited (*p* = 0.05, *p* = 0.05, *p* = 0.01) of ECAR for ATP production in regard to NGF (*p* = 0.0032, *p* = 0.0255, *p* = 0.0460 respectively). TMZ also reduced (*p* = 0.0069, *p* = 0.0006) of ECAR for a non-mitochondrial respiration in regard to PG-1, and NGF respectively. LL-37 reduced (*p* = 0.0033) of ECAR for a non-mitochondrial respiration in comparison with NGF.

### 2.3. Influence of LL-37, PG-1, NGF, and TMZ on Glycolytic Energy Metabolism of Mitochondria in Human Glioma U251 Cells

We measured of extracellular acidification rate for glycolysis, glycolytic capacity and glycolytic spare in glioma U251 cells under the effect of glycose, oligomycin and 2-DG. Besides this basal compounds, we estimated the effect of NGF (7.55 × 10^−3^ µM), LL-37 (4 µM), and TMZ (155 µM) towards the glioma cells, Figure 4.

Results in Figure 4 show that glycolysis, glycolytic capacity and glycolytic spare in glioma U251 cells haven`t been changed under the effect of NGF (7.55 × 10^−3^ µM), LL-37 (4 µM), PG-1 (16 µM) and TMZ (155 µM) in regard to control. TMZ and LL-37 have also inhibited (*p* < 0.0001, *p* = 0.0002) of ECAR for ATP production in comparison with PG-1. However, PG-1 and LL-37 have stimulated of ECAR for ATP production (*p* < 0.0001, *p* = 0.0034) in regard to NGF. This fact indicates that under aerobic conditions, the energy metabolism of tumor cells is realized due to oxidative phosphorylation in mitochondria, while glycolysis is only maintained at a basic level.

### 2.4. Impact of LL-37, PG-1,NGF, and TMZ on the Migration of Human U251 Glioma Cells

We investigated the migration of human glioma U251 cells under the effects of LL-37, NGF, PG-1, and TMZ over 1–6 days (Table 2, Figure 5).

The data in Table 2, Figure 5 show that LL-37 (4.0 μM) stimulated by 2.8 times (*p* = 0.0024) the migration of glioma U251 cells over 1 day (16.8 ± 1.9% versus 6.1 ± 3.6% in control). In contrast, LL-37 inhibited by 3.8 and 3.2 times the migration of glioma cells on the third (6.9 ± 1.1% versus 26.4 ± 12.6% in control, *p* = 0.0065) and fourth days (6.0 ± 3.0% versus 19.1 ± 3.5% in control, *p* = 0.0286). Together, the data in Table 1 and Table 2 demonstrate that LL-37 inhibited the proliferation and clonogenic capability of human glioma U251 cells over 1–2 days and it suppressed the migration of the cells on the third and fourth days. Similarly, NGF (7.55 × 10^−3^ μM) inhibited by 2.3 times (*p* = 0.0159) the migration of human glioma U251 cells over 3–4 days; see Table 2. NGF decreased by 3.1 times the proliferation and clonogenic capability of human glioma U251 cells on the first day; see Table 1. Probably, these data indicate that the effects of LL-37 and NGF involve general intracellular signaling and genetic mechanisms. TMZ inhibited (*p* = 0.0102) the migration of human glioma U251 cells over 3 days; see Table 2.

In contrast, PG-1 (16.0 μM) inhibited by 2.8 times the migration of glioma U251 cells during the first several hours; see Table 2. PG-1 stimulated migration by 2.4 (*p* = 0.0317), 2.5 (*p* = 0.0061), and 3.7 times (*p* = 0.0286) in the glioma cells during the second, fourth, and sixth days, respectively. It is likely that such an effect of PG-1 is associated with the degradation of the peptide containing arginine. Glioma cells, as cancer cells, are arginine-dependent cells. The proliferation and migration of glioma cells are maintained with the intake of arginine; see Table 1 and Table 2. This conclusion is also confirmed by the fact that PG-1 increased the number of cells in the field of view on the second (141 ± 44), third (113 ± 33), fourth (144 ± 37), and sixth days (94 ± 8).

## 3. Discussion

Glioblastoma is the most aggressive brain tumor of the with a median survival rate of 15 months even with aggressive treatment that consists of a combination of surgery, radiation, and chemotherapy. Temozolomide is the standard of care for gliomas, frequently results in resistance to drug and tumor recurrence. Thus, more research is needed for the development of nontoxic and effective drugs in order to guarantee specific treatments to succeed.

We demonstrated that LL-37 (4 μM) and NGF (7.55 × 10^−3^ μM) inhibited the clonogenicity of glioma U251 cells during the first and second days. PG-1 (16 µM) did not impact the clonogenicity of glioma U251 cells over 7 days. Chen X. et al. indicate that human LL-37 inhibits the clonogenicity and proliferation of oral squamous cell carcinoma HSC-3 cells through an increase in the expression of cyclin B1 and PKR-like ER kinase [28]. Inhibition of the p75NGFR receptor suppressed the clonogenicity of non-small cell H460 and lung carcinoma H1299 cells and stimulated their sensitivity to chemotherapy in a mouse xenograft model [29]. Lestaurtinib (K252a—an inhibitor of TrkA receptor) inhibited the formation and the number of colonies of breast cancer MDA-MB-231 cells and the growth of the tumor xenograft in SCID mice [30].

We detected that LL-37 (4 μM) stimulated the migration of glioma U251 cells on the first day. LL-37 and NGF (7.55 × 10^−3^ μM, 100 ng/mL) inhibited the migration of the cells on the third and fourth days. PG-1 (16 µM) inhibited the migration of glioma cells during the first few hours, but it stimulated the migration of the cells during the second, fourth, and sixth days. NGF (200 ng/mL) also inhibited the migration of U87 glioma cells, but it stimulated the migration of SF767 glioma cells via the expression of the p75NTR receptor [31]. The expression of p75NTR induces the stability of hypoxic factors HIF-1α and HIF-2α, which increases the migration, invasion, and stemness of glioma [32]. In contrast, *p75NTR* gene knockdown stimulates the expression of HIF-1α, fibronectin, and L1CAM, and the phosphorylation of Src, focal adhesion kinase (FAK), and paxillin, which increase the migration of C6 glioma cells [33]. NGF stimulates the invasion of breast cancer MDA-MB-231 cells as a result of the binding of CD44 with the TrkA receptor and the activation of the p115RhoGEF/RhoA/ROCK1 cascade [21]. Another possible migration mechanism could be that in which NGF stimulates the assembly of membrane complex TrkA/β1-integrin/FAK/Src and the secretion of matrix metalloprotease-9 [18]. LL-37 inhibits the migration and invasion of HSC-3 cells. In these cells, there is an increase in LL-37 gene expression *CAMP* and activation of caspase-3, with p53-Bcl-2/BAX-dependent apoptosis [28]. In contrast, LL-37 stimulates the internalization of the CXCR4 receptor, activation of the MAPK/Akt/PKC cascade, and migration and invasion of breast cancer MCF7 and MDA-MB-231 cells [26]. In the cancer cells LL-37 can also activate migration via an ErbB2 receptor-dependent mechanism [34]. In A375 and A875 melanoma and squamous cell carcinoma A431 cells, LL-37 (at a dose of 0.05 μg/mL over 24 or 48 h) stimulated migration through the expression of Y-box binding protein-1 (YB-1) and nuclear factor-κB (NF-κB) [35,36]. PG-1 stimulates the migration of IPEC-J2 and DSMZ intestinal cells via the activation of insulin-like growth factor receptor-1 (IGF1R), the expression of C-C motif chemokine ligand-2 (CCL2), prostaglandin-endoperoxide synthase-2 (COX2), interferon regulatory factor 7 (IRF7), epidermal growth factor receptor (EGFR), nuclear factor kappa B (NFKB1), myeloid cell leukemia 1 (MCL1), E-cadherin (CDH1), mucin-1 (MUC1), intercellular adhesion molecule-1 (ICAM1) genes, and the stimulation of the heat shock protein 27 (HSP27)/p38MAPK/extracellular signal-regulated kinase (ERK2)/ cAMP response element-binding protein (CREB) cascade [37].

In this investigation we demonstrated that NGF, PG-1, LL-37, and TMZ inhibited the basal OCR, ATP-synthetase, maximal respiration and activity of cytosolic oxygenases (Figure 3). Arthurs A. et al. [38] studied mitochondrial metabolism in patient-derived U251MG, U373MG, U87MG, T98G, and D54 GBM cell lines and healthy brain tissue. Authors showed that the basal mitochondrial rate and ATP-synthetase were lower than in normal nervous cells. U373MG and T98G cells demonstrated higher basal OCR compared with the primary normal cells. Basal OCR of U87MG and U251MG cells were decreased compared with the primary healthy brain cells. In contrast, the reserve capacity and Krebs cycle capacity were higher in normal brain tissue. Glycolytic capacity is not different between normal and tumor tissues. Investigators concluded that different cell lines display different mitochondrial metabolism compared with normal brain cells. One explanation for this is that tumor cells showed the Warburg effect, reducing their reliance on mitochondrial respiration [39]. Interestingly, that the mitochondrial metabolism of glioblastoma stem cells (GSCs) differs from that of differentiated U87MG glioma cells. Vlashi E. et al. studied the OCR, ECAR, intracellular ATP levels, glucose uptake, lactate production, and radiation sensitivity of GSCs and U87MG, GBM-146, GBM-176 glioma cells [40]. Scientists found GSCs possessed less glucose and lactate while have higher ATP levels than differentiated glioma cells [40]. GSCs were radioresistant and have a higher mitochondrial reserve capacity compared with differentiated cells. Investigators have also assessed can whether the difference between metabolism of GSCs and tumor cells depended on responses to EGF and basic FGF (bFGF) in the media. These growth factors were not affected on the ATP content in populations of GSCs and glioma cells [40].

Gene *PAM16* encodes mitochondria-associated protein involved in granulocyte-macrophage colony-stimulating factor signal transduction. Expression of *PAM16* is effected for cell viability and mitochondrial respiration. This gene is overexpressed in glioma cells of patients with GMB and D-54 MG and U-251 MG glioma cell lines. Inhibition of Magmas decreased basal OCR and ECAR levels in a dose-dependent manner in the D-54 MG and U-251 MG glioma lines [41]. Probably, the inhibitory effects of NGF, LL-37, PG-1 on basal OCR on U251 glioma cells in our experiments can relation to the suppression of *PAM16* overexpression or its protein.

It is well known, that glioma cells overexpressed the epidermal growth factor receptor de2-7EGFR (or EGFRvIII). Low levels of EGFRvIII were also demonstrated in the mitochondria. Activation of Src increased the expression of mitochondrial EGFRvIII. Anna N Cvrljevic et al. showed that low levels of glucose stimulated mitochondrial localization of EGFRvIII in U87MG glioma cells. These cells expressing the EGFRvIII receptors demonstrated increased survival and proliferation. Mitochondrial expression of EGFRvIII inhibited glucose dependency by stimulating mitochondrial oxidative metabolism. Thus, the expression of EGFRvIII in mitochondrial membrane contributes to tumorigenicity glioma cells [42].

## 4. Materials and Methods

### 4.1. Cell Culture

The cell culture of human U251 gliomas was obtained from the Russian Collection of Cell Cultures (Institute of Cytology, Russian Academy of Sciences, Saint-Petersburg, Russia). Human U251 glioma and human tumors cells were cultured at 2.5 × 10^5^/mL in 96-well flat-bottomed plates (TPP, Trasadingen, Switzerland, 1 × 10^4^/well) in Dulbecco’s Modified Eagle Medium DMEM (Sigma-Aldrich, St. Louis, MO, USA) containing 10% fetal bovine serum (Sigma-Aldrich, St. Louis, MO, USA) and gentamicin sulfate 10^−4^ g/mL (Shandong Weifang Pharmaceutical Factory Co., Weifang, China) in a CO_2_ incubator (Heracell, Waltham, MA, USA, 37 °C, 95% humidity and 5% CO_2_) for 1–2 days [43,44]. 

### 4.2. Colony Formation Assay

Human glioma U251 cells were seeded in six-well plates (100 cells on the well) and cultured with NGF, LL-37, PG-1, and TMZ for 7 days. On each day, we photographed the glioma cells with an Axiocam 208 color camera on an inverted microscope Axio vert A1 (Carl Zeiss, Oberkochen, Germany) at objective magnification ×10. We captured 5–10 images in different sectors of each well. We defined the number of colonies and cells in a colony. Every experiment was replicated three times [45].

### 4.3. Scratch Assay

For the study of the migration of glioma U251 cells, we used the 2-well culture insert for self-insertion (lot 200122/2, ibidi GmbH, Gräfelfing, Germany). Each two-well insert was placed into the well of a six-well plate. Human glioma U251 cells were seeded in volumes of 100,000 cells/well at day 0 in a six-well plate [46]. On the day of sowing, in each chamber of the insert with cells, we added 50 μL of LL-37, NGF, PG-1, TMZ or DMEM medium. When the cells sank to the bottom of the well, we removed the inserts. We collected 5–10 images in different areas of the scratch using the Axiocam 208 color camera on the inverted Axio vert A1 microscope (Carl Zeiss, Oberkochen, Germany) at objective magnification ×10 every day for 7 days. We analyzed the images of the cells using ImageJ (version 1.46r, 2012.10.02, National Institute of Mental Health, Bethesda, MD, USA). Wound closure was calculated as the percentage of relation to the scratch created in 0 h.

### 4.4. Seahorse Mito Stress Test Assay

To investigate the effects of mitochondrial inhibitors, also NGF, LL-37, PG-1, and TMZ on mitochondrial respiration in U251 glioma cells we used a Seahorse XF cell mito stress test kit for Agilent Seahorse XF24 analyzer (Seahorse Bioscience, Agilent Technologies, Inc., North Billerica, MA, USA) with according to the manufacture`s protocol [47,48]. Glioma U251 cells were seeded at 2 × 10^4^ cells/well in 96-well Seahorse XF Cell culture miniplate in DMEM with 10% FBS and incubated for 24 h in CO_2_ incubator. Then we diluted glucose up to final 10 mM concentration, oligomycin up to 100 μM, carbonyl cyanide p-trifluoromethoxyphenylhydrazone (FCCP) up to 100 μM and composure of antimycin A plus rotenone up to concentration of 50 μM in Agilent Seahorse XF base medium with 5.5 mM glucose and 2 mM GlutaMAX. The cartridge was hydrated in XF calibrant at 37 °C in a non-CO_2_ incubator overnight. The following day we removed cell media from the plate and washed three times with Seahorse XF base medium. After then we added 100 μL Seahorse phenol red-free DMEM media containing 4 mM L-glutamine (Seahorse Bioscience; Agilent Technologies, Inc., North Billerica, MA, USA). We injected in port A of the Seahorse XF cell culture plate 20 μL oligomycin (1 mM), in port B—22 μL FCCP (0.5 mM), in port C—20 μL of the mixture of antimycin A with rotenone (1 mM) and port D—25 μL NGF in 7.55 × 10^−3^ μM or PG-1 (16 μM), LL-37 (4 μM), and temozolomide (155 μM). The plate was placed in the Seahorse XF96 analyzer which permits to measure oxygen consumption rate (OCR), and extracellular acidification rate (ECAR) or proton production rate in real-time for 90 min (Figure 6).

Addition of oligomycin permits to indicate the level of ATP generation by mitochondria in the glioma cells due to its inhibition of ATP synthase (complex V). FCCP inhibits the oxidative phosphorylation (OXPHOS) in electron-transport chain, that permit to estimate the maximal respiration rate of the cell. Mixture of antimycin A (a complex III inhibitor) with rotenone (a complex I inhibitor) inhibit the mitochondrial respiration completely. This combination lets to measure primarily non-mitochondrial respiration capacity of cytosolic oxidase enzymes [14].

### 4.5. Seahorse Glycolysis Stress Test Assay

To estimate the bioenergy function in U251 glioma cells under the effects of NGF, LL-37, PG-1, and TMZ we used a Seahorse XF cell glycolysis stress test kit for Agilent Seahorse XF24 analyzer (Seahorse Bioscience, Agilent Technologies, Inc., North Billerica, MA, USA) with according to the manufacture`s protocol [35]. Glioma U251 cells were seeded at 2 × 10^4^ cells/well in 96-well Seahorse XF Cell culture miniplate in DMEM with 10% FBS and incubated for 24 h in CO_2_ incubator. Then we diluted in 10-fold times of oligomycin up to final concentration of 100 μM, glucose up to concentration of 100 mM and 2-desoxyglucose (2-DG) up to final concentration of 500 mM. The cartridge was hydrated in XF calibrant at 37 °C in a non-CO_2_ incubator overnight. The next day we removed cell media from the plate and washed three times with Seahorse XF base medium containing with 5.5 mM glucose and 2 mM GlutaMAX. After then was added 100 μL this Seahorse media. We injected in port A of the Seahorse XF cell culture plate of 25 μL NGF in 7.55 × 10^−3^ μM or PG-1 (16 μM), LL-37 (4 μM), and temozolomide (155 μM), in port B – 20 μL glucose (final concentration of 10 mM), in port C—20 μL of oligomycin (1 μM), and port D—27 μL 2-DG (50 mM). Agilent Seahorse XF24 analyzer permits to estimate the basal level of medium acidification, glycolysis medium acidification, maximum glycolytic capacity, glycolytic reserve and non-glycolytic medium acidification by using extracellular acidification rate (ECAR), Figure 7 [49].

Oligomycin inhibits of ATP synthase (complex V) and increases ECAR level that permit to study the level of ATP generation by mitochondria in human U251 glioma cells. 2-DG is an inhibitor of hexokinase of glycolysis that lets to measure the rate of non-glycolytic acidification.

### 4.6. Reagents

Human cathelicidin LL-37 (AnaSpec, Fremont, CA, USA); porcine protegrin-1 (PG-1), (SynPep, Dublin, CA, USA); 7S of nerve growth factor from murine submaxillary gland (Sigma-Aldrich, Darmstadt, Germany); gentamicin sulfate (solution for intravenous and intramuscular administration, 40 mg/mL, Shandong Weifang Pharmaceutical Factory Co., Liaocheng, China); temozolomide (Temodal capsules, 100 mg, Orion Corporation, Orion Pharma, Espoo, Finland); an XF Cell Mito Stress Test Kit and Glycolysis Stress Test kit were purchased from Seahorse Bioscience (USA).

### 4.7. Statistical Analysis

All experiments were performed at least in triplicate. The statistical significance of the differences between the means of different treatments and their respective control groups was determined using Student’s t-test. Data were counted with the standard deviation and considered significant at *p* < 0.05. To compare the differences between two independent groups with a small number of samples (*n* < 30), the nonparametric Mann–Whitney U-test was used [50]. The descriptive statistics were performed using the GraphPad Prism software (version 6.01, 09.21.2012, Dotmatics, San Diego, CA, USA).

## 5. Conclusions

LL-37, NGF, and TMZ inhibit migration and clonogenicity of U251 glioma cells, which may indicate that these compounds have anti-mitogenic and anti-migration effects on human glioma cells. NGF, PG-1, LL-37, and TMZ inhibited the basal OCR, ATP-synthetase, maximal respiration and activity of cytosolic oxygenases. These substances hasn’t changed spare respiration capacity of mitochondria. Also, TMZ, LL-37, and PG-1 decreased of ECAR for a non-mitochondrial respiration in regard to control level. Glycolysis, glycolytic capacity and glycolytic spare in glioma U251 cells haven`t been changed under the effect of NGF, LL-37, PG-1, and TMZ in regard to control level. Anticancer activities of NGF, and LL-37, PG-1 maybe is relation to their capacity to reduce the basal OCR, ATP-synthetase, maximal respiration of mitochondria in human glioma U251 cells. The study of the mechanisms of these effects may contribute in the future to the use of NGF and LL-37 as therapeutic agents for gliomas.

## Figures and Tables

**Figure 1 molecules-27-04988-f001:**
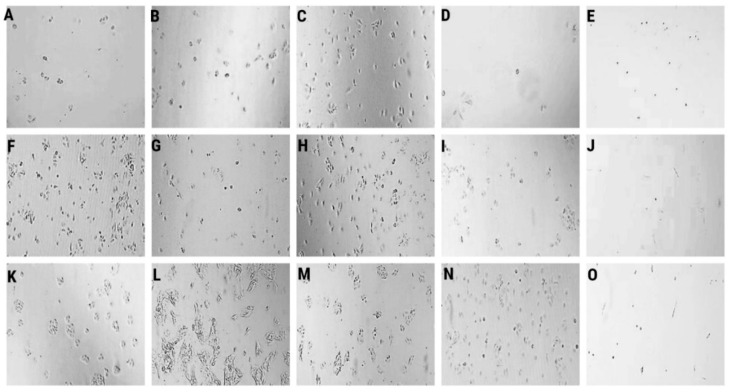
Representative microscopy images of the colony formation assays are shown. Clonogenic capability of human glioma U251 cells over 1 day in control (**A**), LL-37 (4.0 μM) (**B**), PG-1 (16.0 μM) (**C**), NGF (7.55 × 10^−3^ μM) (**D**), TMZ (155 μM) (**E**), over 2 day in control (**F**), LL-37 (4.0 μM) (**G**), PG-1 (16.0 μM) (**H**), NGF (7.55 × 10^−3^ μM) (**I**), TMZ (155 μM) (**J**), over 3 day in control (**K**), LL-37 (4.0 μM) (**L**), PG-1 (16.0 μM) (**M**), NGF (7.55 × 10^−3^ μM) (**N**), TMZ (155 μM) (**O**). Lens magnification ×10.

**Figure 2 molecules-27-04988-f002:**
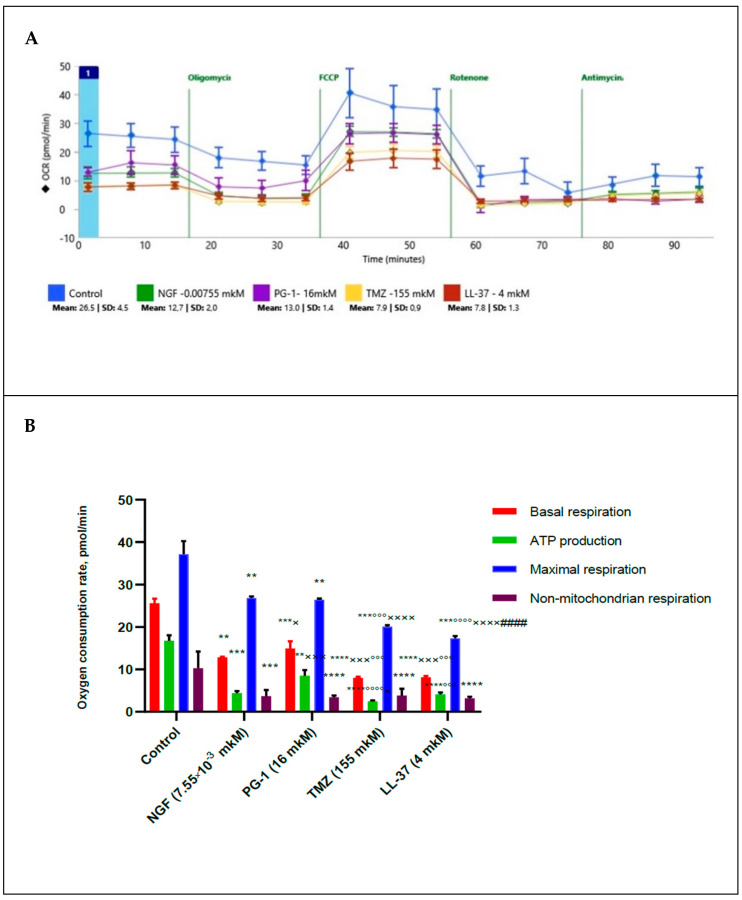
Oxygen consumption rate in human glioma U251 cells under the effect of NGF (7.55 × 10^−3^ µM), LL-37 (4 µM), PG-1 (16 µM), and TMZ (155 µM) over time (**A**); in comparison to control with reagents (**B**): **—significance (*p* < 0.01), *** (*p* < 0.001), **** (*p* < 0.0001) from control; ×—significance (*p* ≤ 0.05), ××× (*p* < 0.001), ×××× (*p* < 0.0001) from NGF; °°°—significance (*p* < 0.001), °°°° (*p* < 0.0001) from PG-1; ####—significance (*p* < 0.0001) from TMZ.

**Figure 3 molecules-27-04988-f003:**
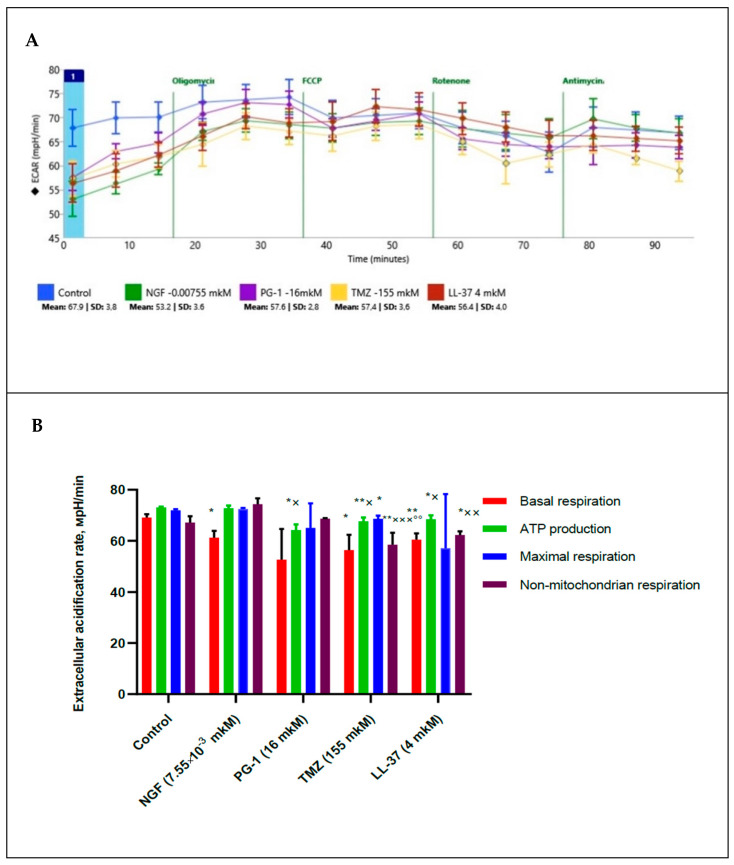
Extracellular acidification rate (ECAR) in human U251 glioma cells under the effect of NGF (7.55 × 10^−3^ µM), LL-37 (4 µM), PG-1 (16 µM), and TMZ (155 µM) over time (**A**); in comparison to control with reagents (**B**): *—significance (*p* ≤ 0.05), ** (*p* < 0.01) from control; ×—significance (*p* ≤ 0.05), ×× (*p* < 0.01), ××× (*p* < 0.001) from NGF; °°—significance (*p* < 0.01) from PG-1.

**Figure 4 molecules-27-04988-f004:**
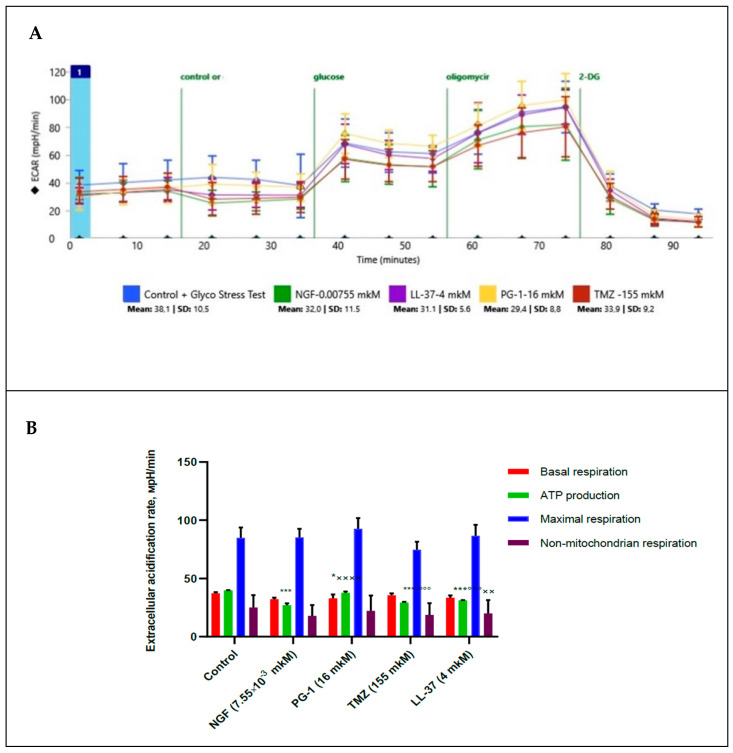
Extracellular acidification rate (ECAR) in human U251 glioma cells under the effect NGF.(7.55 × 10^−3^ µM), LL-37 (4 µM), PG-1 (16 µM) and TMZ (155 µM) over time (**A**); in comparison to control with reagents (**B**). *—significance (*p* ≤ 0.05), *** (*p* < 0.001) from control; ××—significance (*p* < 0.01), ×××× (*p* < 0.0001) from NGF; °°°—significance (*p* < 0.001), °°°° (*p* < 0.0001) from PG-1.

**Figure 5 molecules-27-04988-f005:**
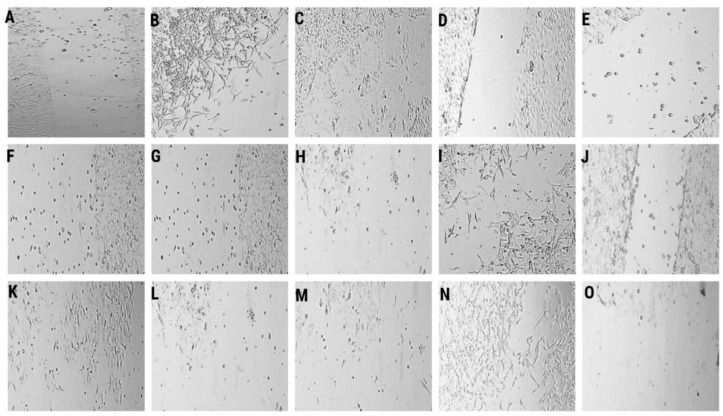
Representative microscopy images of transwell migration of U251 glioma cells over 1 day: control (**A**), under the action of NGF (7.55 × 10^−3^ μM) (**B**), LL-37 (4 μM) (**C**), PG-1 (16 μM) (**D**), and TMZ (155 μM) (**E**); over 2 days: control (**F**), under the action of NGF (7.55 × 10^−3^ μM) (**G**), LL-37 (4 μM) (**H**), PG-1 (16 μM) (**I**), and TMZ (155 μM) (**J**); over 3 days: control (**K**), under the action of NGF (7.55 × 10^−3^ μM) (**L**), LL-37 (4 μM) (**M**), PG-1 (16 μM) (**N**), and TMZ (155 μM) (**O**). Magnification ×100.

**Figure 6 molecules-27-04988-f006:**
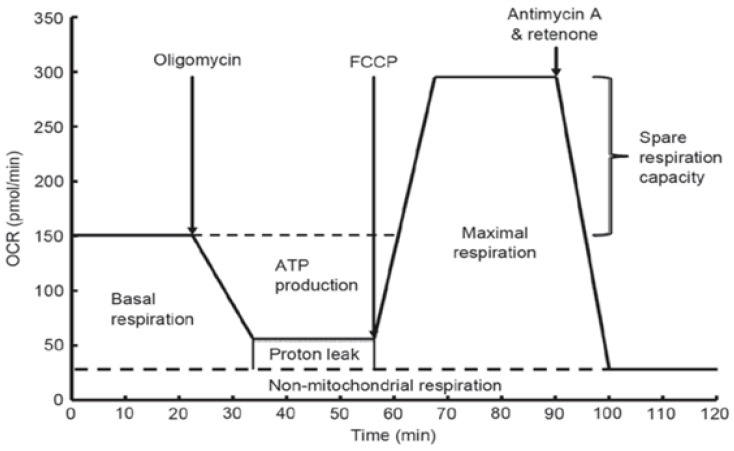
Evaluation of various types of mitochondrial respiration by oxygen uptake rate using the Seahorse Bioscience XF24 analyzer.

**Figure 7 molecules-27-04988-f007:**
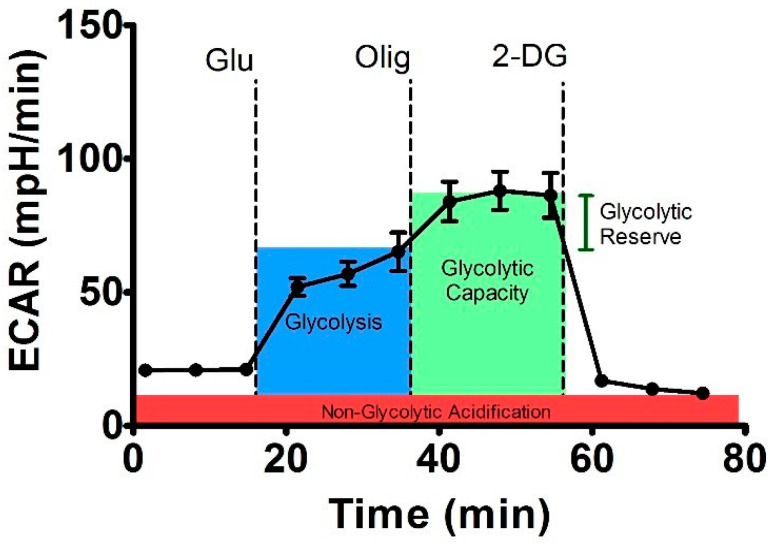
Scheme of the experiment for assessing the glycolytic capacity of the cell using the analyzer XF24 Seahorse Bioscience [49].

**Table 1 molecules-27-04988-t001:** Impact of LL-37, PG-1, and NGF on clonogenicity of U251 glioma cells.

Day	Parameter	Control	LL-37	PG-1	NGF	TMZ
1 day	The percentage of cells in the clone to the total number	73.8 ± 18.1	**29.3 ± 18.9 *°** **(*p* = 0.05)**	77.2 ± 15.5	**25.0 ± 9.3 *°** **(*p* = 0.0152)**	**13.8 ± 2.9 **°** **(*p* = 0.001)**
	The percentage of inhibition of the cell growth	-	**60.3 ± 5.1 *** **(*p* = 0.05)**		**66.2 ± 3.7 *** **(*p* = 0.0152)**	**81.3** **± 5.5 *×†** **(*p* = 0.001)**
	Average number of cells per field of view	17.0 ± 11.0	6.0 ± 2.0	12.0 ± 4.0	**5.0 ± 3.0 *** **(*p* = 0.05)**	**29.5 ± 6.4 *×†** **(*p* = 0.010)**
	Number/percentage of single cells	3.0 ± 2.0/17.6	4.0 ± 2.0/**66.7 ***	3.0 ± 2 /25.0	**3.0 ± 1.0 */** **60.0 ***	**100 *×°†**
2 days	The percentage of cells in the clone to the total number	96.1 ± 6.1	**53.9 ± 23.5 ***** **(*p* = 0.0001)**	68.7 ± 28.6	**75.6 ± 8.8 *##** **(*p* = 0.05)**	**10.0 ± 2.2 **××†** **(*p* = 0.001)**
	The percentage of inhibition of the cell growth	-	**43.9 ± 4.3 ***** **(*p* = 0.0001)**	28.5	21.4	**73.5** **± 3.9 **†×°** **(*p* = 0.001)**
	Average number of cells per field of view	23.0±15.0	17.0 ± 5.0	11 ± 7	10.0 ± 6.0	**25.5 ± 10.0 *** **(*p* = 0.010)**
	Number/percentage of single cells	2.0 ± 1.0/ 8.6	**6.0 ± 3.0 */35.3 *#** **(*p* = 0.0364)**	3.0 ± 2.0 / **27.3 *#**	**2.0 ± 1.0 */** **20.0 *#**	**100 *°×†**
3 days	The percentage of cells in the clone to the total number	91.0 ± 15.7	83.8 ± 11.4 ##	87.1 ± 15.3 ##	90.4 ± 8.3 ##	**8.7 ± 1.8 **††××°°** **(*p* = 0.001)**
	The percentage of inhibition of the cell growth	-	7.9	4.3	0.7	**74.2 ± 4.6 **×°†** **(*p* = 0.001)**
	Average number of cells per field of view	59.0 ± 10.0	42.0 ± 12.0	43.0 ± 25	25.0 ± 13.0	**23.5 ± 4.9 *** **(*p* = 0.01)**
	Number/percentage of single cells	4.0 ± 2.0/ 6.8	6.0 ± 3.0/ 14.3 ##	4.0 ± 2/9.3 ##	2.0 ± 0.6/ 8.0 ##	**100 ± 0.0 *†°×**
6 days	The percentage of cells in the clone to the total number	99.0 ± 0.9	89.8 ± 4.4	88.0 ± 13.8	95.9 ± 4.2	-
	The percentage of inhibition of the cell growth	-	9.3	11.2%	3.2	-
	Average number of cells per field of view	105.0 ± 44.0	**65.0 ± 20.0 *** **(*p* = 0.0444)**	110.0 ± 69.0	**50.0 ± 10.0 *** **(*p* = 0.0238)**	-
	Number/percentage of single cells	2.0 ± 0/1.9	**6.0 ± 3.0 */9.2** **(*p* = 0.0285)**	**5 ± 3 * /4.5** **(*p* = 0.0160)**	2.0 ± 1.0/4.0	-
7 days	The percentage of cells in the clone to the total number	97.4 ± 1.8	91.7 ± 6.1	98.8 ± 1.0	95.7 ± 1.7	-
	The percentage of inhibition of the cell growth	-	5.9	-	1.7	-
	Average number of cells per field of view	148.0 ± 61.0	118.0 ± 46.0	200.0 ± 59.0	**63.0 ± 18.0 *** **(*p* = 0.0191)**	-
	Number/percentage of single cells	6.0 ± 1.0/4.0	8.0 ± 5.0/6.8	**2.0 ± 1.0 */1.0** **(*p* = 0.0286)**	**1.0 ± 0.7 */1.6** **(*p* = 0.0286)**	-

Note: Significant results are shown in bold face. Data are expressed as mean ± standard deviation. Each time point is represented by the average value obtained from the analysis of 5 photos. *—significance (*p* ≤ 0.05), ** (*p* < 0.01), *** (*p* < 0.001) from control; × (*p* < 0.05) from NGF, ×× (*p* < 0.01) from NGF; °—significance (*p* ≤ 0.05), °° (*p* < 0.01) from PG-1; #—significance (*p* ≤ 0.05), ## (*p* < 0.01) from TMZ; †—significance (*p* ≤ 0.05) from LL-37; †† (*p* ≤ 0.01) from LL-37.

**Table 2 molecules-27-04988-t002:** Quantitation of human glioma U251 cells migration. The wound healing rate, %.

Day	The Wound Healing Rate,% The Average Number of Cells, Peptides, Growth Factor, Dose, μM				
	Control	LL-37 (4.0)	PG-1 (16.0)	NGF (7.55 × 10^−3^)	TMZ (155.0)
0 day	3.9 ± 1.3 (43 ± 25)	6.8 ± 1.9 (44 ± 13)	**1.4 ± 0.5 *×#** (9 ± 3) ***p* = 0.05**	4.1 ± 1.9 (36 ± 24)	5.4 ± 0.7 (86 ± 31)
1 day	6.1 ± 3.6 (42 ± 27)	**16.8 ± 1.9 **×#** (87 ± 44) ***p* = 0.0024**	8.7 ± 4.1 (50 ± 29)	7.6 ± 2.9 (62 ± 27)	7.8 ± 3.3 (62 ± 14)
2 days	8.2 ± 5.4 (79 ± 34)	5.9 ± 0.4 (59 ± 25)	**19.8 ± 10.2 *×#†** (141 ± 44) ***p* = 0.0317**	7.6 ± 0.4 (59 ± 7)	6.6 ± 1.7 (55 ± 21)
3 days	26.4 ± 12.6 (97 ± 9)	**6.9 ± 1.1 **°#** (77 ± 21) ***p* = 0.0065**	31.7 ± 18.5 † (113 ± 33)	**11.4 ± 3.5 *** (87 ± 25) ***p* = 0.0159**	**8.9 ± 1.9 *** (70 ± 19) ***p* = 0.0102**
4 days	19.1 ± 3.5 (61 ± 7)	**6.0 ± 3.0 *°×** (48 ± 22) ***p* = 0.0286**	**48.6 ± 14.6 **** (144 ± 37) ***p* = 0.0061**	**11.4 ± 1.8 *†°** (84 ± 27) ***p* = 0.0159**	9.7 ± 2.8 (92 ± 23)
6 days	13.8 ± 1.3 (101 ± 13)	16.8 ± 7.2 (47 ± 19)	**51.7 ± 7.1 *†** (94 ± 8) ***p* = 0.0286**	-	-

Note: Significant results are shown in bold face. Data are expressed as mean ± standard deviation. Each time point is represented by the average value obtained from the analysis of 5 photos. *—significance (*p* ≤ 0.05), ** (*p* < 0.01) from control; ×—significance (*p* ≤ 0.05), from NGF; #—significance (*p* ≤ 0.05) from TMZ, †—significance (*p* ≤ 0.05) from LL-37, °—significance (*p* ≤ 0.05) from PG-1.

## Data Availability

The data presented in this study are openly available on Figshare at 10.6084/m9.figshare.16879432.

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
