# Peer review of "Anticancer Effect of Cathelicidin LL-37, Protegrin PG-1, Nerve Growth Factor NGF, and Temozolomide: Impact on the Mitochondrial Metabolism, Clonogenic Potential, and Migration of Human U251 Glioma Cells"

_molecules, 2022, doi:10.3390/molecules27154988_

Round 1
Reviewer 1 Report
The paper in my opinion cannot be accepted in the present form.
In this paper authors evaluate the cytotoxic effects of LL37,PG1 and NGF in human U251 glioma cells.
The papers includes a lot of data, in my opinion, organized in a chaotic manner , difficult to be understood . I should suggest to reorganize the data concerning LL37,PG1 and NGF effects on U251 cells ,compared with chemotherapy treatment, matching them, in the same figures and tables or in different figures and tables, with LL37,PG1 and NGF effects on U251 cells ,compared with chemotherapy treatment on astrocytoma ( Fig.7-8,Tab.4) and glioblastoma cells (Fig.9-10,Tab.5).
The remaining data, in my opinion, could be used for another paper.
Specific issues.
Authors should be know that MTT assay evaluates cell metabolic activity and not the cytotoxic effect: each legend related to MTT assay data should be changed. LDH assay should be more addressed to evaluate cytotoxic effect.
-Why the same scale is not used in all figures?
-Why the legends to all figures do not include standard deviation and number of replicated samples?
Fig.8 Authors state that this graph shows viability instead of cytotoxic effect as well as Fig.6,for example, both according to MTT assay. What does it mean? Why authors use different term since both data seem to be acquired according to MTT?
In MM section line 480 C 6 instead of U251 cells seems to be erronously reported.
Why authors did not compare the effects of LL37,PG1 and NGF as well as on migration with those of chemoterapeutic agents.
Conclusions are not expressed in this manuscript.
Author Response
Dear Reviewer 1,
Thank you very much for a thorough analysis of the manuscript and valuable comments.
We made the following changes according to the reviewer's comments:
- The papers includes a lot of data, in my opinion, organized in a chaotic manner , difficult to be understood . I should suggest to reorganize the data concerning LL37,PG1 and NGF effects on U251 cells ,compared with chemotherapy treatment, matching them, in the same figures and tables or in different figures and tables, with LL37,PG1 and NGF effects on U251 cells ,compared with chemotherapy treatment on astrocytoma ( Fig.7-8,Tab.4) and glioblastoma cells (Fig.9-10,Tab.5).
The remaining data, in my opinion, could be used for another paper.
We reorganized the data concerning LL37, PG1, NGF, and TMZ effects on U251 cells. We have included the analysis of the mitochondrial metabolism in glioma cells in study of the clonogenicity and migration of U251 glioma cells.
- Why authors did not compare the effects of LL37,PG1 and NGF as well as on migration with those of chemoterapeutic agents.
Since, Temozolomide is the standard of care for gliomas, frequently results in resistance to drug and tumor recurrence, we added the analysis of effect of TMZ on the clonogenicity and migration of human glioma U251 cells as well as the mitochondrial metabolism in glioma cells in manuscript.
- Conclusions are not expressed in this manuscript.
We added Conclusion in this manuscript.
I thank you very much for your interest and I look forward to hearing from you.

Reviewer 2 Report
The manuscript presented to MOLECULES is of average interest.
Similar studies have been performed in vitro on U251 cells studying synergistic effects. Here are three examples:
Journal of Oncology
https://www.spandidos-publications.com/10.3892/ijo.2021.5241
Cell Death and Disease
https://www.nature.com/articles/s41419-021-03535-9
Molecules (MDPI)
https://www.mdpi.com/1420-3049/27/4/1299
The Abstract is too long and does not follow the same structure. For example, Glioblastoma is abbreviated (GBM) but nerve growth factor (NGF) is not. The Abstract is confusing as to what is the goal of the study. There is too much information and not a final conclusion of what the study has achieved.
The Conclusion is another reflection of this idea. The Conclusion focuses again on results. Discuss again the best combinations of treatments and the effects. There is no conclusion. There is no future work delineated or what are some implications of the research team's findings. As mentioned above, this type of experimentation has been done before, extensively for anything that is cancer-related.
FOr example, in the MOLECULES article mentioned above, the article concludes:
Our study might encourage clinical trials focusing on the evaluation of the safety, tolerability and preliminary efficacy of peptide nucleic acid administered to patients, as in the case of NCT05212532 and NCT00127517, based on a PNA (EOM613/AVR 118) exhibiting immunomodulatory, antiviral and anticancer properties [86]. In this respect, several clinical studies based on SFN administration to cancer patients have been proposed, such as NCT03232138 (lung cancer), NCT01228084 (prostate cancer), NCT00982319 (breast cancer) and NCT01568996 (melanoma).
The methods are polarized toward Cytotoxicity. This is appropriate yet makes the manuscript less attractive. Figure 11 seems to lack controls. All the samples and the index calculated are treatments. The explanation of this particular Index in Figure 11 is confusing.
Finally, U251 is not the only cell line. GBM cell lines (LN229, SNB19, U87, U251) are another example. Selecting the best synergistic treatment may be tested on a separate GBM cell line or tumorigenic cells with a similar genetic background.
I believe the work was done correctly, the discussion and conclusion are not sound. This work does not compare to others of similar scientific motivation, including the above-mentioned Molecules 2022, 27(4), 1299; https://doi.org/10.3390/molecules27041299.
This could be more of an Editorial decision as to the interest of Molecules to publish for the readership, and to maintain the Impact Factor.
Author Response
Dear Reviewer 2,
Thank you very much for a thorough analysis of the manuscript and valuable comments.
We made the following changes according to the reviewer's comments:
- The methods are polarized toward Cytotoxicity. This is appropriate yet makes the manuscript less attractive.
We reorganized the data concerning LL37, PG1, NGF, and TMZ effects on U251 cells. We have included the analysis of the mitochondrial metabolism in glioma cells in study of the clonogenicity and migration of U251 glioma cells.
- The Abstract is too long and does not follow the same structure.
We made the following changes in the Abstract.
- The Conclusion is another reflection of this idea. The Conclusion focuses again on results. Discuss again the best combinations of treatments and the effects. There is no conclusion. There is no future work delineated or what are some implications of the research team's findings.
We added the Conclusion in this manuscript.
We thank you very much for your interest and we look forward to hearing from you.

Round 2
Reviewer 2 Report
No additional comments
Author Response
Dear Reviewer 2,
Thank you very much for a thorough analysis of the manuscript and valuable comments.
We made the following changes according to the reviewer's comments:
- English language and style are fine/minor spell check required
Manuscript underwent extensive English revisions by MDPI Editing Services
- Are the results clearly presented?
We reorganized the data presentation
We thank you very much for your interest and we look forward to hearing from you.